# Oncologic Outcome of Robotic-Assisted and Laparoscopic Sentinel Node Biopsy in Endometrial Cancer

**DOI:** 10.3390/cancers15245894

**Published:** 2023-12-18

**Authors:** Atanas Ignatov, József Mészáros, Stylianos Ivros, Paolo Gennari, Tanja Ignatov

**Affiliations:** 1Department of Gynecology and Obstetrics, Otto-von-Guericke University, 39108 Magdeburg, Germany; jozsef.meszaros@med.ovgu.de (J.M.); stylianos.ivros@med.ovgu.de (S.I.); paolo.gennari@med.ovgu.de (P.G.); dr.t.ignatov@ivf-muenchen.de (T.I.); 2Gynecologic Oncology Unit, Metropolitan Hospital, 18547 Athens, Greece

**Keywords:** endometrial cancer, sentinel node, lymph-node dissection, survival

## Abstract

**Simple Summary:**

In our study of 419 patients with endometrial cancer, we looked at two ways of checking the spread of the disease: one called ‘sentinel lymph node biopsy’ (SLNB) and the other, a more traditional method called ‘lymph node dissection’ (LND). We followed these patients for about 5.5 years. We found that both methods had similar outcomes in terms of how long patients lived without the disease coming back (known as disease-free survival) and how long they lived overall (overall survival). Whether patients had their complete lymph nodes checked or only the sentinel did not seem to change how well they did. But we did notice that the kind of treatment they received afterward could affect how they did. Overall, it seems like the newer SLNB method is just as good as the traditional LND in helping patients with endometrial cancer.

**Abstract:**

Background: Recently, sentinel lymph node biopsy (SLNB) has been introduced in the surgical staging of endometrial cancer as an alternative to systematic lymph node dissection (LND). However, the survival impact of SLNB is not yet well characterised. Methods: We performed a retrospective study of 419 patients with endometrial cancer treated with SLNB alone or with pelvic and para-aortic LND. For SLNB mapping, indocyanine green was used. Results: Median follow-up was 66 months. After exclusions, 337 patients were eligible for analysis. Of them, 150 underwent SLNB and 187 LND. During the follow-up time, 27 (24.7%) of the 150 who underwent SLNB and 54 (28.9%) of the 187 who underwent LND were diagnosed with recurrent disease (*p* = 0.459). The estimated 5-year disease-free survival (DFS) rate was 76.7% and 72.2% for patients in the SLNB and LND group, respectively (*p* = 0.419). The 5-year overall survival (OS) rates were 80.7% and 77.0% in the SLNB and LND group, respectively (*p* = 0.895). Survival rates were similar in both groups independent of lymph node status. Multivariable analysis confirmed that the staging approach was not associated with oncological outcome. For patients without lymph node metastases, patient outcome was worsened by advanced tumour stage and non-endometrioid tumour histology. In the group of patients with confirmed lymph node metastases, advanced tumour stage and inadequate adjuvant treatment significantly reduced DFS and OS. Conclusion: Our data suggested that SLNB did not compromise the oncological outcome of patients with endometrial cancer compared to LND.

## 1. Introduction

Endometrial cancer is the most common gynaecologic cancer worldwide and surgical staging involves the assessment of pelvic and para-aortic lymph node status [1]. However, the survival benefit of lymph node dissection (LND) remains controversial. To date, two randomised prospective trials investigating the benefits of systematic LND compared to no staging have failed to show any improvement in a patient’s survival [2,3]. These results are in contrast to retrospective series, which suggest a survival benefit to systematic LND [4,5,6,7]. The uncertain survival benefit of LND and the high rate of complications have led to the development of alternative approaches, such as sentinel lymph node biopsy (SLNB) [8]. This technique involves identifying and examining the sentinel lymph node(s), which are the initial lymph nodes to receive drainage from the tumour site. It has been shown that SLNB is non-inferior to systematic LND in detecting lymph node metastases. Compared to traditional systematic LND, SLNB is associated with lower rates of complications [9,10,11,12,13]. It minimizes the need for extensive lymph node removal, potentially reducing the risk of postoperative complications. There is a growing body of evidence suggesting that SLNB is safe and feasible in both low- and high-risk endometrial cancer patients [14,15,16,17,18,19,20,21]. Its potential to provide accurate staging with reduced invasiveness marks it as a significant area of interest in improving the care and outcomes for individuals diagnosed with endometrial cancer.

However, the effect of SLNB on the oncologic outcome of patients with endometrial cancer is still unknown, and the question of whether patients with positive sentinel lymph node(s) should be treated by systematic LND has also yet to be clarified [8,22]. Management protocols for patients with positive sentinel nodes require clarification. The question of whether these patients should undergo further lymph node dissection or receive alternative treatments remains unresolved.

Here, we aim to compare survival of patients receiving SLNB with those undergoing LND in this real-world group of patients with endometrial cancer. In the group of patients treated with SLNB, systemic LND was not performed even where metastases were found in sentinel lymph node(s).

## 2. Materials and Methods

### 2.1. Patients

In this particular cohort study, all individuals who received surgical staging for confirmed endometrial cancer were enrolled. These patients were treated at the University Hospital Magdeburg, Germany, spanning from January 2012 to December 2020. Excluded from the study were patients with initial metastatic disease or those who did not undergo lymph node staging. The eligible female participants were categorized into two groups based on the method of staging employed: (1) LND and (2) SLNB. All participants underwent a surgical staging procedure that involved total hysterectomy and bilateral salpingo-oophorectomy. If the histological type was non-endometrial, an omentectomy was additionally performed, and peritoneal samples were obtained. In the LND group, patients received systematic pelvic and para-aortic LND following previously established protocols [1]. In the SLNB group, patients underwent an SLNB algorithm [1]. In this group, no LND was undertaken when one or more positive sentinel lymph nodes were found. Adjuvant therapy was evaluated and compared between groups. The manuscript was prepared in accordance with the STROBE statement criteria [23].

### 2.2. SLNB and LND

Indocyanine green (ICG) was used for SLNB and the mapping was performed as already described [1]: indocyanine green (ICG) was employed for SLNB according to the described protocol. A solution containing 25 mg of ICG powder dissolved in 20 mL of sterile water was gradually injected into the cervical mucosa and submucosa at the 3 and 9 o’clock positions. This injection took place in the operating room while the patient was under anaesthesia. Near-infrared fluorescence was used to assess ICG-mapped sentinel lymph nodes, enabling the visualization of lymph vessels and nodes in a green hue. The sentinel lymph node, identified as the first juxtauterine fluorescence-positive node, was documented and subsequently removed. In cases where bilateral mapping failed, an option was either a re-injection of ICG or a LND on the contralateral hemipelvis. These cases were included into the SLNB group as described previously [1]. The SLNB procedure was conducted using conventional laparoscopy or robotic-assisted minimally invasive surgery, with subsequent histopathological assessment of the excised lymph nodes.

Systematic LND involved the removal of pelvic and para-aortic lymph nodes up to the renal vessels, followed by histopathological evaluation. Patients at low risk of recurrence were exempt from undergoing LND. The definition of the prognostic risk groups was in accordance with ESGO/ESTRO/ESP guidelines [24]. Lymph node status was classified based on absence of tumour infiltration (reported as negative), presence of micrometastases (tumour in lymph node sized >0.2 mm and ≤2.0 mm), or identification of macrometastases (>2.0 mm) [1]. Notably, the preoperative imaging and the presence of suspicious lymph node involvement as well as myometrium invasion were not considered in the decision-making process.

### 2.3. Statistical Analysis

Statistical analyses were conducted using SPSS version 28.0 (SPSS, Chicago, IL, USA). Group comparisons for clinical, pathological, and treatment parameters were performed using appropriate tests for categorical and continuous variables. Survival probabilities were estimated through the Kaplan-Meier method, and the equality of survival curves was assessed using the log-rank test. Disease-free survival (DFS), defined as the time between diagnosis and the occurrence of local/regional recurrence, distant metastases, or disease-related death, was the primary outcome measure. Overall survival (OS), from diagnosis to death from any cause, was the secondary outcome. Follow-up extended until the patient’s death, the last available information, or the final follow-up as of 9 March 2023. Statistical significance was set at *p*-values less than 0.05 in the two-sided analyses.

## 3. Results

The median follow-up was 66 months (range 2–146 months). Over the study period 419 patients with endometrial cancer were treated. Of them, 82 were excluded from this analysis (Figure 1), because of metastatic disease (*n* = 35) or missing lymph node staging (*n* = 47).

Thus, 337 women were eligible for analysis: 150 had undergone SLNB and 187 LND. Most of the clinical and pathological parameters were equally distributed between the two groups (Table 1). However, in the SLNB group, significantly more patients had low-grade tumours (68.9%) compared to the LND group (47.6%). This is because LND was not performed in low-risk patients. Patients who underwent LND were more likely to have a myometrium invasion of more than 50%, compared with the patients who underwent SLNB. The median number of extirpated lymph nodes was significantly lower in the SLNB group (median 3, range 2–27) compared to the LND group (median 42, range 6–88) (Table 1, *p* < 0.0001).

The recurrence rate was equally distributed between the two groups (Table 2; *p* = 0.459). Twenty-seven (24.7%) of 150 patients and 54 (28.9%) of 187 women in the SLNB and LND group, respectively, were diagnosed with recurrent disease during the follow-up period. The rates of pelvic, para-aortic, and distant lymph node recurrence were similar in the SLNB and LND groups.

Regarding survival outcome, survival curves suggested that DFS was similar in the two groups (Figure 2A, *p* = 0.386). The estimated 5-year DFS rate was 76.7% and 72.2% for patients in the SLNB and LND groups, respectively. During the follow-up, 33 (22.0%) and 48 (25.7%) deaths were recorded in the SLNB and LND groups, respectively. The 5-year OS rates were 80.7% and 77.0% in the SLNB and LND group, respectively (Figure 2B, *p* = 0.482).

Survival outcomes did not differ significantly between the treatment groups when subdivided by lymph node status. As shown in Table 1, 27 (18.0%) of 150 women and 31 (16.6%) of 187 women had positive lymph nodes in the SLNB and LND groups, respectively. The 5-year DFS rates were 34.4% for patients treated with SLNB and 35.5% for patients treated with LND in the group of patients with positive lymph nodes (Figure 3A). This was not statistically different between the groups (*p* = 0.676). The 5-year OS was also similar between the two groups (Figure 3B, *p* = 0.608). The estimated 5-year OS rates were 37.0% and 48.4% for the SLNB and LND groups, respectively. Similarly, in the group of patients with negative lymph nodes, the staging approach did not influence the outcome. The 5-year DFS was 83.7% and 79.5% in SLNB and LND group, respectively, (Figure 3C, *p* = 0.361). The 5-year OS was 90.2% and 82.7% in SLNB and LND groups, respectively, (Figure 3D, *p* = 0.097).

To investigate the factors associated with the DFS and OS, a multivariable analysis was performed (Table 3). In the group of patients without lymph node metastases, patient outcomes were worsened by advanced tumour stage for both DFS (HR = 4.56; 95% CI 1.64–12.64; *p* = 0.004) and OS (HR = 3.07; 95% CI 1.04–9.02; *p* = 0.041). Non-endometrioid tumour histology was also associated with worse DFS (HR =2.70; 95% CI 1.30–5.61; *p* = 0.008) and OS (HR = 2.11; 95% CI 1.07–4.15; *p* = 0.031). In the group of patients with lymph node metastases, the tumour stage was also an unfavourable prognostic factor for DFS (HR = 3.80; 95% CI 1.61–8.95; *p* =0.002) and OS (HR = 3.01; 95% CI 1.23–7.32; *p* = 0.016). Adjuvant radiotherapy and chemotherapy significantly positively influenced DFS (HR = 0.22; 95% CI 0.07–0.64; *p* = 0.006) and OS (HR = 0.18; 95% CI 0.06–0.55; *p* = 0.003) but not radiotherapy alone (HR = 0.42; 95% CI 0.10–1.69). Notably, the staging approach was not associated with survival independent of lymph node status.

## 4. Discussion

Since the incorporation of SLNB in the staging of endometrial cancer in the early 2000s, its utilisation has increased exponentially. The accuracy of SLNB as a diagnostic approach has been demonstrated for patients with low as well as high risk of recurrence [8,22]. However, the survival outcomes of SLNB are still unknown and a topic of ongoing debate. In the present study, we were able to demonstrate that SLNB in endometrial cancer was not negatively associated with survival outcomes compared with the standard procedure LND.

Recent retrospective studies have been concordant with our findings and showed non-inferiority of SLNB in patients with endometrial cancer with different characteristics including: low-risk endometrial cancer [22]; endometrioid endometrial cancer with >50% myometrial invasion [25]; carcinosarcoma [26]; and serous and clear cell carcinoma [27,28]. In contrast, our cohort was heterogeneous and represented the real-word cohort of patients with endometrial cancer. The rate of patients with type II endometrial cancer was estimated to be 18% of the whole cohort. In this context, our data demonstrated that SLNB is usable and safe for all types of endometrial cancer.

However, although the therapeutic effect of LND has not been confirmed in prospective randomised trials [2,3], the most important critique of SLNB is the failed removal of possible non-sentinel metastatic lymph nodes. Inadequate para-aortic staging is another point of contention. In our cohort, the oncologic outcome was not compromised by SLNB independently of lymph node status. Similarly, a retrospective study found that SLNB and LND had similar survival outcomes in patients with endometrioid endometrial cancer with >50% myometrial invasion [25]. The same research group confirmed this finding in a cohort of patients with serous and clear cell carcinoma [28]. Furthermore, Basaran et al. reported that the omission of para-aortic LND did not affect the survival effect of SLNB as staging approach [27]. There are two plausible explanations of our results. Firstly, using ultrastaging, significantly more micrometastases in pelvic lymph nodes can be diagnosed [29], reducing the rate of isolated para-aortic lymph node metastases by up to 1% [30]. Furthermore, the rate and pattern of recurrence observed by us was similar between the LND and SLNB groups and is comparable with the data reported by others [31]. For example, the observed recurrences in the regional and para-aortic and/or distant lymph nodes were similar in the LND and SLNB groups. These results suggest no therapeutic effect of systemic LND as already observed in two randomised trials [2,3]. On the other hand, the negative survival effect of lymph node metastases could be neutralised by adjuvant treatment. We demonstrated that radiotherapy and chemotherapy were highly predictive of improved survival for patients with lymph node metastases, but not for lymph node negative patients. It has been also found by multivariable analysis that adjuvant treatment was a significant predictor of OS in the whole cohort but not in the subgroup of patients with negative or unknown lymph node status alone [25,26,28]. However, this comparison should be interpreted with caution, due to the different rates of adjuvant therapy received in the LND and SLNB groups. More patients in the SLNB cohort received adjuvant therapy [25,26,28]. The follow-up periods were also different in the SLNB and LND group. In our study, the rates of adjuvant treatment and the follow-up period were comparable between the SLNB and LND groups. Furthermore, our findings, particularly regarding the benefit of adjuvant treatment for patients with positive lymph nodes, are supported by results from the PORTEC-3 trial [32] in which chemotherapy plus radiation improved DFS in patients with high-risk endometrial cancer.

These findings support the importance of adjuvant treatment, particularly in the group of patients with lymph node metastases. The SLNB approach is associated with the detection of more lymph node metastases and subsequently upstaging and appropriate adjuvant management. However, the Endometrial Cancer Lymphadenectomy Trial (ECLAT) may answer the question as to whether the removal of metastatic lymph nodes has a therapeutic effect [33]. In most published studies evaluating the oncologic impact of SLNB, where positive sentinel lymph node(s) were found, a LND was subsequently performed and the real oncological impact of SLNB alone could not be investigated. In our cohort, LND was not performed in the SLNB group even in the case of metastatic sentinel lymph node(s) being found on SLNB, giving us real information about the survival effect of sentinel mapping compared with the standard LND. After adjustment for clinical and pathological variables, patient survival was not affected by the staging approach used, suggesting that LND possibly had no additional therapeutic effect.

An important limitation of our study is its retrospective monocentric nature. Nevertheless, the data were prospectively evaluated, resulting in high completeness of the relevant clinical and pathological data. Surveillance was performed as suggested by existing guidelines resulting in a long follow-up, which is comparable in both groups. A further limitation is the lack of randomisation of the staging approach used. Thus, significantly more patients with low-differentiated tumours were treated by SLNB. A multivariate analysis was undertaken to reduce the impact of selection bias. Moreover, adjuvant treatment was administered per the existing guidelines and the use of radiotherapy and chemotherapy equally distributed between LND and SLNB groups.

A significant strength of our study is that in the group of SLNB patients, no LND was subsequently performed even in the case of positive sentinel(s). In this way, we were able to compare the survival of the whole LND and SLNB groups, not only in patients with negative lymph nodes. The study population was similar to the general population including all risk groups for endometrial cancer and the exclusion criteria were kept to a minimum. Thus, we obtained a cohort with high level of external validity. In contrast to some aforementioned studies, the rate of adjuvant treatment in our cohort was similar between both groups. Moreover, patients treated by SLNB and LND were treated during the same period, suggesting that adjuvant management during the time period was similar. Furthermore, this study has a long follow-up period compared to other published studies on this topic.

## 5. Conclusions

The findings indicate that the innovative SLNB technique does not compromise the outcomes for individuals battling endometrial cancer when compared to the conventional LND method. Both SLNB and LND showed comparable effectiveness regarding patient outcomes. This suggests that the newer SLNB approach could be a viable and safe alternative to the established LND method in staging endometrial cancer.

Moreover, the study’s comprehensive analysis, encompassing factors like tumour stage, histology, and adjuvant treatments, highlighted key influencers in patient outcomes. While advanced tumour stage and non-endometrioid tumour histology impacted outcomes in patients without lymph node metastases, those with confirmed lymph node involvement faced challenges related to advanced tumour stage and inadequate adjuvant treatments. This underlines the importance of tailoring treatments based on specific patient factors rather than solely relying on the staging approach. Overall, the study encourages confidence in the efficacy of SLNB, offering a potentially less invasive yet equally effective alternative for endometrial cancer staging while emphasizing the critical role of personalized treatments in improving patient outcomes.

## Figures and Tables

**Figure 1 cancers-15-05894-f001:**
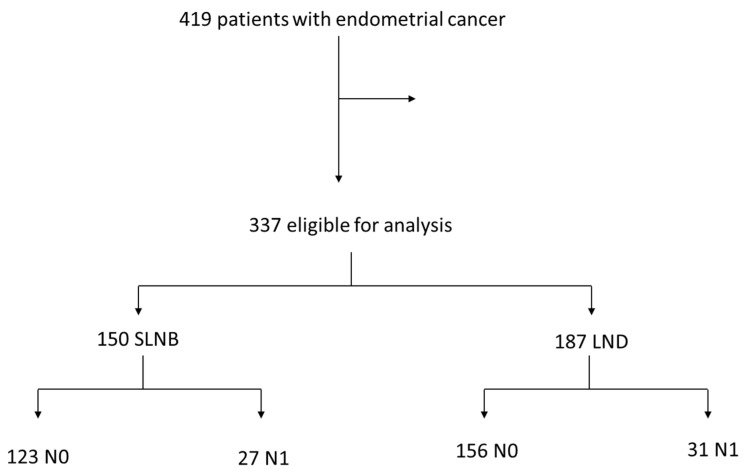
Study design.

**Figure 2 cancers-15-05894-f002:**
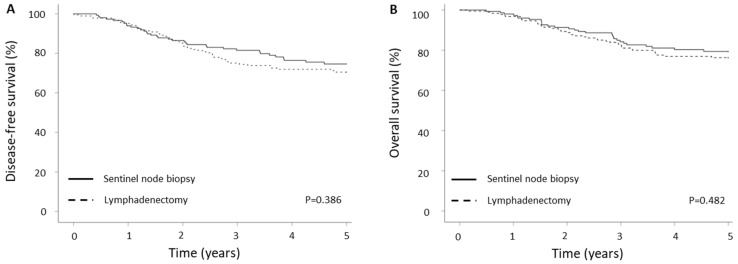
Disease-free (**A**) and overall survival (**B**) by staging procedure.

**Figure 3 cancers-15-05894-f003:**
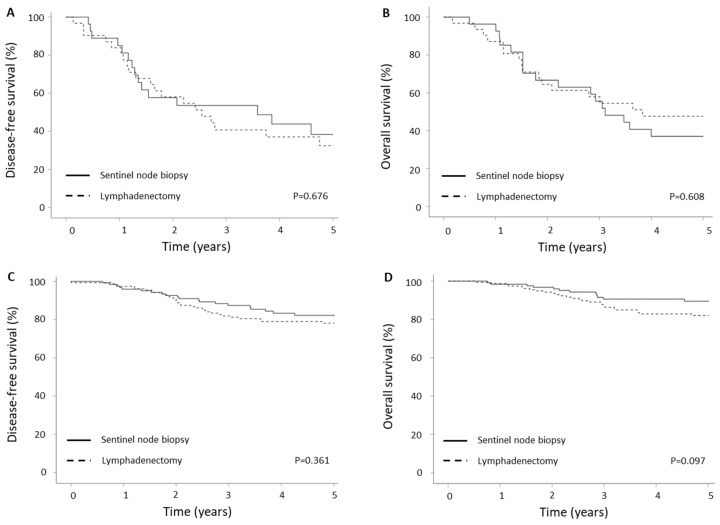
Survival outcomes depending on lymph node status. (**A**) Disease-free and (**B**) overall survival of patients with lymph node metastases. (**C**) Disease-free and (**D**) overall survival of patients without lymph node metastases.

**Table 1 cancers-15-05894-t001:** Clinical and pathological parameters.

	SnB	LAD	*p*-Value
Total	150 (44.5%)	187 (55.5%)	
Age, median	67 (42–86)	67 (33–81)	0.804
Stage			
I	114 (76.0%)	127 (67.9%)	0.217
II	15 (10.0%)	29 (15.5%)	
III	21 (14.0%)	31 (16.6%)	
Grading			
Low	102 (68.9%)	89 (47.6%)	<0.001
High	46 (31.1%)	98 (52.4%)	
Myometrium invasion			
<50%	70 (49%)	49 (27.5%)	<0.001
>50%	73 (51.0%)	129 (72.5%)	
LVSI			
Negative	110 (73.8%)	39 (26.2%)	0.393
Positive	125 (69.1%)	56 (30.9%)	
Extirpated LN, median	3 (2–27)	42 (6–88)	<0.001
LN-Status			
Negative	123 (82.0%)	156 (83.4%)	0.772
Positive	27 (18.0%)	31 (16.6%)	
Adjuvant Therapy			
No	86 (57.3%)	96 (51.3%)	0.748
Radiotherapy	17 (11.3%)	25 (13.4%)	
Chemotherapy	10 (6.7%)	14 (7.5%)	
Both	37 (24.7%)	52 (27.8%)	

**Table 2 cancers-15-05894-t002:** Pattern of recurrence.

	SnB	LND	*p*-Value
Total	37 (24.7%)	54 (59.3%)	0.459
Local	14 (9.3%)	12 (6.4%)	0.412
Regional lymph nodes	5 (3.3%)	11 (5.9%)	0.313
Paraaortic/distant lymph nodes	4 (2.7%)	7 (3.7%)	0.760
Abdominal	8 (5.3%)	9 (4.8%)	1.000
Distant	12 (8%)	25 (13.4%)	0.160

**Table 3 cancers-15-05894-t003:** Multivariable analysis of DFS and OS.

Variable	Patients with Negative Lymph Nodes				Patients with Positive Lymph Nodes			
	DFS		OS		DFS		OS	
	HR, (CI 95%)	*p*-Value	HR, (CI 95%)	*p*-Value	HR, (CI 95%)	*p*-Value	HR, (CI 95%)	*p*-Value
Approach		0.172		0.058		0.257		0.496
SLNB	Reference		Reference		Reference		Reference	
LND	1.75 (0.79–3.90)		2.47 (0.97–6.30)		1.16 (0.71–3.68)		1.39 (0.54–356)	
Age	0.97 (0.94–1.01)	0.142	0.99 (0.96–1.04)	0.868	0.97 (0.93–1.000)	0.052	0.99 (0.96–1.02)	0.544
Histology	c	0.008		0.031		0.327		0.902
Type I	Reference		Reference		Reference		Reference	
Type II	2.70 (1.30–5.61)		2.11 (1.07–4.15)		1.35 (0.74–2.49)		0.95 (0.45–2.00)	
Stage		0.004		0.041		0.002		0.016
FIGO I/II	Reference		Reference		Reference		Reference	
FIGO III	4.56 (1.64–12.64)		3.07 (1.04–9.02)		3.80 (1.61–8.95)		3.01 (1.23–7.32)	
Grading		0.093		0.47		0.65		0.739
Low	Reference		Reference		Reference		Reference	
High	0.39 (0.13–1.17)		0.69 (0.25–1.90)		0.77 (0.24–2.41)		0.83 (0.27–2.56)	
LVSI		0.528		0.382		0.774		0.807
Negative	Reference		Reference		Reference		Reference	
Positive	0.72 (0.26–1.98)		1.53 (0.59–3.94)		1.17 (0.40–3.41)		0.88 (0.30–2.56)	
Adjuvant therapy		0.103		0.631		0.006		0.003
None	Reference		References		Reference		Reference	
RT	2.06 (0.96–3.41)		0.86 (0.32–2.27)		0.44 (0.10–2.26)		0.42 (0.10–1.69)	
RT + CT	2.28 (0.85–6.16)		1.28 (0.46–3.56)		0.22 (0.07–0.64)		0.18 (0.06–0.55)	

## Data Availability

The data presented in this study are available on request from the corresponding author. The data are not publicly available due to ethical restrictions.

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
