# Peer review of "Oncologic Outcome of Robotic-Assisted and Laparoscopic Sentinel Node Biopsy in Endometrial Cancer"

_cancers, 2023, doi:10.3390/cancers15245894_

Round 1

Reviewer 1 Report

Comments and Suggestions for Authors

In table 1, SnB, LAD are should be corrected into SLNB, LND.

Comments on the Quality of English Language

English is fine.

Author Response

Table 1 was changed as suggested.

Reviewer 2 Report

Comments and Suggestions for Authors

Very interested article, despite the retrospective design.

Is stratification based on histological subtype available?

There are no results for histological type II in the multivariate analysis.

Author Response

We greatly appreciate these kind comments. The histological type was included in the multivariable analysis and is displayed in Table 3. We presume the reviewer might not have noticed it.

Reviewer 3 Report

Comments and Suggestions for Authors

The introduction of indocyanine green (ICG) mapping sentinel lymph node biopsy as an alternative to systematic lymph node dissection in the surgical staging of endometrial cancer is a noteworthy development. Previous studies have shown that SLNB is comparable to systematic LND in detecting lymph node metastases while exhibiting a lower incidence of complications in both low- and high-risk endometrial cancer. However, the impact of SLNB on survival outcomes remains a subject necessitating further elucidation. Within this retrospective study encompassing patients with endometrial cancer subjected to either with SLNB alone or with pelvic and para-aortic LND, the findings concluded that SLNB aligns with systematic LND in terms of oncological outcomes. This study significantly contributes to our understanding of the oncologic outcomes in endometrial cancer patients undergoing sentinel node biopsy.

 This manuscript is well-structured, with clear and concise figures that contribute to a comprehensive understanding of the presented data. However, there are certain aspects where additional clarification and elaboration would greatly enhance the overall depth of the paper. The following are my questions and suggestions regarding this manuscript.

1.     Regarding the Materials and Methods section, specifically patient selection, are there factors that influence whether a patient undergoes systematic lymph node dissection (LND) or sentinel lymph node biopsy (SLNB)? In the article, it is mentioned that patients with a low risk of recurrence will undergo sentinel lymph node biopsy (SLNB). Is there a clearly defined criteria or definition for low risk of recurrence?

2.     Before surgery, is an imaging examination routinely conducted, and do the results of such imaging influence the decision between performing sentinel lymph node biopsy (SLNB) or systematic lymph node dissection (LND)? Additionally, does the identification of myometrium invasion of one half or more or the presence of lymphadenopathy in imaging studies impact this decision?"

3.     In the case of unilateral hemi-pelvis failed mapping necessitating unilateral LND, would this be categorized as part of the LND group or the SLNB group?

4.     If a suspicious node is identified during sentinel lymph node biopsy (SLNB), would the course of action involve not only removing the suspicious node but also transitioning to systematic lymph node dissection (LND)?

5.     In Table 1, Clinical and pathological parameters, providing more detailed histology types in the cancer grading section and information on the presence or absence myometrium invasion would enhance the comprehensiveness of the data.

Author Response

  1. Regarding the Materials and Methods section, specifically patient selection, are there factors that influence whether a patient undergoes systematic lymph node dissection (LND) or sentinel lymph node biopsy (SLNB)? In the article, it is mentioned that patients with a low risk of recurrence will undergo sentinel lymph node biopsy (SLNB). Is there a clearly defined criteria or definition for low risk of recurrence?

There were no specific factors that influenced patient selection for the use of SLNB and LND. Importantly, patients at a low risk of recurrence, for whom LND was not recommended, underwent SLND. This decision stemmed from the understanding that even patients at low-risk could potentially have lymph node metastases. The definition of low-risk recurrence patients adhered to the ESGO/ESTRO/ESP guidelines (Concin et al., Int J Gynecol Cancer, 2021). This is mentioned in the new version of the manuscript (line 102).

  1. Before surgery, is an imaging examination routinely conducted, and do the results of such imaging influence the decision between performing sentinel lymph node biopsy (SLNB) or systematic lymph node dissection (LND)? Additionally, does the identification of myometrium invasion of one half or more or the presence of lymphadenopathy in imaging studies impact this decision?"

Before surgery, staging procedures were conducted. The results of such imaging did not influence the decision to perform an LND or SLNB. In other words, the presence of lymphadenopathy or suspicion of lymph node invasion, as well as myometrium invasion, did not affect the decision-making process. To make it clear for the readership we included following sentence (line 106):

Notably, the preoperative imaging and the presence of suspicious lymph node involvement as well as myometrium invasion were not considered in the decision-making process.

  1. In the case of unilateral hemi-pelvis failed mapping necessitating unilateral LND, would this be categorized as part of the LND group or the SLNB group?

As outlined in the 'Patients and Methods' section, when bilateral mapping proved unsuccessful, the available options included either a re-injection of ICG or performing a LND on the contralateral hemipelvis (line 95-96). In response to the reviewer's suggestion and to ensure clarity for the readers, we have added:

“These cases were included into the SLNB group as described previously (Ignatov et al)”

which is detailed in line 96."

  1. If a suspicious node is identified during sentinel lymph node biopsy (SLNB), would the course of action involve not only removing the suspicious node but also transitioning to systematic lymph node dissection (LND)?

The reviewer's opinion aligns with current standards and guidelines, which is valid.

Yet, it's essential to note that in our specific study group, a distinct feature was the absence of lymph node dissection (LND) despite the detection of one or more positive sentinel lymph nodes. This distinct aspect is highlighted in line 82.

  1. In Table 1, Clinical and pathological parameters, providing more detailed histology types in the cancer grading section and information on the presence or absence myometrium invasion would enhance the comprehensiveness of the data.

In the grading representation, we used the new classification, namely tumors with differentiation grade 1 or 2 were referred to as 'low,' and tumors with differentiation grade 3 as 'high’.

As suggested by the reviewers, the myometrium invasion was analyzed and is present in the revised version of the manuscript (s. Table 1). This new information is discussed in the main text in line 132. The reviewer is correct in her/his opinion that this information will enhance the comprehensiveness of our data.